# Residential Food Environment, Household Wealth and Maternal Education Association to Preschoolers’ Consumption of Plant-Based Vitamin A-Rich Foods: The EAT Addis Survey in Addis Ababa

**DOI:** 10.3390/nu14020296

**Published:** 2022-01-11

**Authors:** Adane Kebede, Magnus Jirström, Alemayehu Worku, Kassahun Alemu, Hanna Y. Berhane, Christopher Turner, Eva-Charlotte Ekström, Yemane Berhane

**Affiliations:** 1Department of Health System and Policy, Institute of Public Health, College of Medicine and Health Sciences, University of Gondar, Gondar P.O. Box 196, Ethiopia; 2Department of Human Geography, Lund University, 223 62 Lund, Sweden; magnus.jirstrom@keg.lu.se; 3School of Public Health, College of Health Science, Addis Ababa University, Addis Ababa 1176, Ethiopia; alemayehuwy@yahoo.com; 4Department of Epidemiology and Biostatistics, Institute of Public Health, College of Medicine and Health Sciences, University of Gondar, Gondar P.O. Box 196, Ethiopia; kassalemu@gmail.com; 5Department of Nutrition and Behavioral Sciences, Addis Continental Institute of Public Health, Addis Ababa 26751/1000, Ethiopia; hannayaciph@gmail.com; 6Department of Women’s and Children Health, Uppsala University, 751 85 Uppsala, Sweden; Lotta.Ekstrom@kbh.uu.se (E.-C.E.); yemaneberhane@gmail.com (Y.B.); 7Department of Population Health, Faculty of Epidemiology and Population Health, London School of Hygiene and Tropical Medicine, Keppel Street, London WC1E 7HT, UK; C.J.Turner@greenwich.ac.uk; 8Department of Epidemiology and Biostatistics, Addis Continental Institute of Public Health, Addis Ababa 26751/1000, Ethiopia

**Keywords:** vitamin A-rich vegetables, vitamin A-rich fruits, food availability, vendors, residential, food environment

## Abstract

Vitamin A deficiency is common among preschoolers in low-income settings and a serious public health concern due to its association to increased morbidity and mortality. The limited consumption of vitamin A-rich food is contributing to the problem. Many factors may influence children’s diet, including residential food environment, household wealth, and maternal education. However, very few studies in low-income settings have examined the relationship of these factors to children’s diet together. This study aimed to assess the importance of residential food availability of three plant-based groups of vitamin A-rich foods, household wealth, and maternal education for preschoolers’ consumption of plant-based vitamin A-rich foods in Addis Ababa. A multistage sampling procedure was used to enroll 5467 households with under-five children and 233 residential food environments with 2568 vendors. Data were analyzed using a multilevel binary logistic regression model. Overall, 36% (95% CI: 34.26, 36.95) of the study children reportedly consumed at least one plant-based vitamin A-rich food group in the 24-h dietary recall period. The odds of consuming any plant-based vitamin A-rich food were significantly higher among children whose mothers had a higher education level (AOR: 2.55; 95% CI: 2.01, 3.25), those living in the highest wealth quintile households (AOR: 2.37; 95% CI: 1.92, 2.93), and in residentials where vitamin A-rich fruits were available (AOR: 1.20; 95% CI: 1.02, 1.41). Further research in residential food environment is necessary to understand the purchasing habits, affordability, and desirability of plant-based vitamin A-rich foods to widen strategic options to improve its consumption among preschoolers in low-income and low-education communities.

## 1. Introduction

Vitamin A deficiency (VAD) is a serious public health problem and one of the nutritional challenges faced by children in low-income countries, especially among those from the poorest communities [1]. Pre-school children are the most vulnerable population group to VAD [2], due to the relatively high demand for vitamin A during this period of rapid growth and development [2]. VAD in this age group is aggravated by low micronutrient intake and high incidence of infections [3,4]. VAD affects approximately a third of all preschool children in the world [5]. Preschoolers in sub-Saharan African countries (SSA) are disproportionately affected by VAD, with nearly half (48%) of all preschoolers suffering from VAD [5]. In 2017, the prevalence of VAD in Ethiopia was 28%, featuring the second-highest prevalence of VAD among preschoolers in the world [6]. Over 95% of the world’s deaths attributable to VAD occurred in sub-Saharan Africa (SSA) and south Asia; this deficiency also accounted for 2.0% of all deaths in sub-Saharan preschoolers [5]. In Ethiopia, the all-age total DALYs due to VAD were 397.8 years in thousand [6]; the prevalence of night blindness (2.8%) and Bitot’s spots (2.1%) among preschool children are higher than the WHO cutoff point [7]. The trend of VAD in Ethiopia, SSA, and across the globe from 1990 to 2017 showed decreasing and consistently higher in Ethiopia compared to in the SSA and the globe [6]. In Ethiopia, the consumption of vitamin A-rich fruits and vegetable in children is low, accounting for less than 10% of total weight consumed [8].

Although vitamin A supplementation (VAS) has been implemented via routine child health programs in an effort to address VAD in Ethiopia [9,10,11], supplementation alone cannot be a sustainable long-term solution [12]. Food-based solutions targeting improved dietary diversity among vulnerable populations provide pathway to the long-term prevention of VAD [13,14].

The food environment is increasingly recognized to play significant role in shaping the diet of the children and their families globally [15]. A food environment which includes availability, accessibility, affordability, desirability, convenience, marketing, and properties of food products is the space where children and their families interact with the wider food system for food acquisition and consumption [16]. Foods that are commonly available in the residential food environment have been shown to influence the eating behaviors of families and their children [17]. In Ethiopia, the limited affordability of vitamin A source foods groups in urban residential food environments was associated with lower intake of fruit and vegetables [18,19]. Improving availability of vitamin A-rich fruits, dark green leafy vegetables, and vitamin A-rich vegetables and roots is a food based strategy aiming to increase the consumption of carotene rich fruit and vegetables and to combat VAD [20,21]. However, the food environment is not well researched in low-income settings [22]. Especially in rapidly growing cities in Africa, such as Addis Ababa, the residential environment is constantly changing due to development projects and resettlement of residents. This has a potential to disrupt the residential food environment, which may also affect food availability and consumption [23].

A systematic review of evidence in low-and middle-income countries has shown the influence of household socio-economic status on fruit and vegetable intake [24]. A recent study in Ghana and India found that vitamin A-rich fruit and vegetable consumption was higher among children belonging to the richest household wealth strata and having mothers with higher education [25,26]. On the other hand, studies from the Philippines and Brazil discovered that a low intake of fruit and vegetables among children in the richest household [27,28].

Understanding the importance of food availability for preschoolers’ diet and comparing its significance with that of commonly addressed socio-economic characteristics can contribute with a more in-depth understanding of the factors which need to be addressed to promote healthy dietary habits.

This study aims to examine the consumption level of plant-based vitamin A-rich foods among children under the age of five, and to assess to what extent the residential availability of vitamin A-rich plant-based foods, household wealth, and maternal education influence the consumption of these foods by children aged 6 to 59 months in Addis Ababa.

## 2. Materials and Methods

### 2.1. Study Design and Setting

Addis Ababa is the largest city in Ethiopia, and one of the fastest-growing cities in the African continent [29], with an estimated population of 3.6 million, 10.6% of which are estimated to be children under five years of age [30]. The population highly heterogeneous with regard to economic status [31], with an unemployment rate of 31.4%, and 18.9% of the population living below poverty line. The city is also the largest urban recipient of migrants [32,33]. Together, the services (63%) and industry (36%) sectors share almost all economic structure of the city [33]. The food environment in Addis Ababa is diverse, with vendors ranging from micro-vendors (locally known as Gulit) to formal supermarkets [34].

At the time of data collection, Addis Ababa was administratively divided into 10 sub-cities and 117 woredas (the smallest formal administrative unit), with each sub-city comprising of 10–15 woredas.

This study utilizes data collected in the EAT Addis survey which collected data from households and vendors in residential areas in all woredas of Addis Ababa. Data collection was conducted in two rounds to account for seasonality, capturing July to August 2017, reflecting a wet season, and January to February 2018, reflecting the post-harvest period.

### 2.2. Sampling and Eligibility

We used a multistage sampling procedure. Each of the 117 woredas in Addis Ababa were included in the survey. Each woreda was divided into five clusters to simulate the enumeration procedure in the Ethiopian Demographic and Health Surveys [30]. One cluster from each woreda was selected for inclusion using a simple random sampling procedure.

To identify eligible households with children under five years of age in the cluster, we used a systematic random sampling procedure visiting every third household from a random starting position until we reached a total of 60 households per cluster. These households were assessed for eligibility based on presence of at least one child under five years of age. In cases where an eligible household had more than one eligible child, one child was randomly selected to be the index child from whom dietary data was to be collected. The mother or caretaker of the index child was invited to participate in the survey and was also the main respondent at the household data collection. Mothers or caregivers not at home following three recruitment visits were declared unavailable. For the purpose of this study, household with children age below six months were not included.

To identify a residential food environment representing the cluster, one household per cluster was randomly selected to serve as an index household, around which all vendors within a five-minute walk in all direction were surveyed. In total, 14 type of vendors such as kiosk, micro vendor, bakery, fruit/vegetable shop, four mill, butcher, cooperative shop, ET-Fruit, street food vendor, mini market, dairy shop, mobile micro-vendor, livestock market, and fish market were included in the survey. Vendors which were not open at the time of the visit were declared unavailable.

### 2.3. Data Collection and Measurement

Instruments were developed to collect data from households and vendors. These instruments were drafted in English and subsequently translated into Amharic. All data collectors and supervisors had extensive previous field research experience and received training on the data collection tools, interview procedures, and ethical conduct pertaining to the study [35]. Pilot testing refined the development of our survey tools and procedures, including the use of tablets with Open Data Kit (ODK) software. Primary data was sent directly to a protected data server at Addis Continental Institute of Public Health.

#### 2.3.1. Food Groups

An overall aim in the EAT Addis survey was to evaluate the association between food availability in residential food environments and food consumption of the household and pre-school children. To facilitate comparison, we developed a common metric to assess food availability and food consumption. A frequently used type of indicator is diet diversity, and indicators have been developed for use in different population groups including children [36,37,38]. We developed a set of food groups which could be used to derive diet diversity indicators both for the household and the preschoolers, as well as to be used to define availability in the residential area. For this particular study we used collected information on three plant-based vitamin A-rich food groups: vitamin A-rich fruits, dark green leafy vegetables, and vitamin A-rich vegetables and roots.

#### 2.3.2. Household Survey

The household survey featured data collection on a number of aspects, but of relevance for this study is mainly infant and young child feeding, household wealth, maternal education, and distance between households and index household.

Children aged 6–59 months were included in this study. The caregiver of the index child was asked to recall the child’s food consumption during 24 h recall period. In addition to the initial recall by the caretaker a photo gallery of locally available items was used to augment the listing of food groups to ensure a common understanding among our enumerators and participants. For the purpose of this study, three photos with common items representative of the food groups were used: vitamin A-rich vegetables (pumpkin, carrot, red bell pepper), vitamin A-rich fruits (mango, papaya), and dark green leafy vegetables (amaranth, cassava leaves, Ethiopian kale, cabbage, Swiss chard, broccoli). Consumption was categorized as yes or no to each of the three food groups.

The household wealth index was developed by principal component analysis and categorized into wealth quintiles [39]. This was calculated based on household assets, household characteristics, access to utilities, and infrastructure variables. Maternal education was assessed based on the reported highest level of grade completed by mothers at the time of the survey, and then categorized according to the Ethiopian educational system—never attend school/not finished first grade, grade 1–4, grade 5–8, grade 9–12, and college-educated [40]. Food security was assessed by use of household food insecurity access scale [41].

Distance between households and index household were calculated based on geographical positioning system (GPS coordinates) of each household during the data collection. The Euclidian distance was calculated based on the geographical location points (latitude/longitude) using a trigonometric approach (distance formula) [42].

#### 2.3.3. Residential Food Environment Survey

A survey of all food vendors within a five-minute walking radius from the index household was conducted. The data collection tool for the residential food environment survey consists of the vendor characteristics, vendor properties, and food availability. The survey tool was pilot tested in three randomly selected residential food environments to assess the feasibility and the relevance of the terminology and typologies. The same food groups and photo gallery of locally available food items used in the household survey was also used in the survey of vendors in the residential area. Of relevance for this study, each vendor was categorized as selling or not selling any items from each of the three vitamin A food rich food groups, such as vitamin A-rich fruits, dark green leafy vegetables, and vitamin A-rich vegetables and roots. For each food group, the residential availability was defined as the presence of at least one vendor in the area selling any item from that particular group [43].

### 2.4. Statistical Analysis

We used STATA 14 software for data analysis. Frequencies and percentages were calculated to report descriptive statistics. The predictor of plant-based vitamin A-rich food consumption was assessed using a multilevel binary logistic regression analysis with meaningful nested hierarchy at household and residential food environment level [44]. The intra-cluster correlation coefficient (ICC) in an empty model showed variability in child consumption of plant-based vitamin A-rich food attributed to differences in residential food environment, also referred as between-cluster variability. Initially, each independent variable was evaluated individually to generate unadjusted effect estimates. After this, three multivariable multilevel logistic regression models were fitted. Model I included residential food environment variables (level 2 variables), comprised of availability of vitamin A-rich vegetables and roots, dark green leafy vegetables, and vitamin A-rich fruits. Model II included household and individual variables (level 1 variables), comprised of household and individual status. Thus, Model I adjusted for residential wealth in tertial and household distance to the indexed household, and Model II adjusted for maternal age, child age, marital status, and number of under-five children [45]. Model III included both level 2 and level 1 variables together to adjust for maternal age, child age, marital status, number of under-five children, household distance to the indexed household, and residential wealth tertial. The observed associations were expressed as unadjusted and adjusted odds ratio with 95% confidence intervals.

### 2.5. Ethical Consideration

This study was approved by the Ethical Review Board of Addis Continental Institute of Public Health (Ref No. ACIPH/IRB/004/2015), and the Ethical Review Board at University of Gondar (R.No.-V/P/RCS/05/355/2019). No identifying information was available for this study.

## 3. Results

A total of 14,018 households were visited, leading to the identification of 5467 eligible and consenting households. Among eligible households, 4911 children were in the age group 6 to 59 months and were eligible for inclusion in this study, while the remaining 556 children were under 6 month of age and therefore ineligible for this study, as exclusive breast feeding is recommended in this age group (Figure 1A). In the food environment survey, we identified 2680 vendors in 233 residential food environments, of which we managed to include 2568 vendors; 112 food vendors were unavailable at the time of the survey (Figure 1B).

### 3.1. Characteristics of the Respondents and Study Children

Male and female children were almost equally represented in the sample in all age groups. The mean (±SD) age of the children was 28.4 (±0.2) months. The majority (88%) of the children’s mothers were married. Approximately two-thirds (74%) of the mothers reported no income-earning activities. Approximately one-fifth of the mothers (19%) had a college education, an access of private toilet (23%), and lived in their own house (22%) (Table 1)

### 3.2. Children’s Consumption of Plant-Based Vitamin A-Rich Foods

Consumption of plant-based vitamin A-rich food was limited, with only 36% of the children having consumed at least one type of vitamin A-rich food in the previous 24 h. The highest consumed vitamin A-rich food item was vitamin A-rich vegetables and roots (20%), followed by dark green leafy vegetables (17%) and vitamin A-rich fruits (9%). Only 9% of the children had consumed more than one type of plant-based vitamin A-rich foods in the previous 24 h.

### 3.3. Residential Availability of Plant-Based Vitamin A-Rich Food Groups

The majority (78%) of residential food environments had at least one of the plant-based vitamin A-rich food available. The residential availability of vitamin A-rich fruits was low (36%), as compared with the other two vitamin A-rich food items (Table 2). The most commonly available vitamin A-rich plant sources in the residential food environment were vegetables and roots.

### 3.4. Consumption of Plant-Based Vitamin A-Rich Foods among Children by Residential Food Availability

Children residing in residential food environments where dark green leafy vegetables were available had significantly higher consumption of vitamin A-rich rich plant source foods (37%), compared to those residing in residential food environments where these foods were not available (34%) (*p*-value = 0.03). Similarly, children residing in residential food environments where vitamin A-rich fruits were available had significantly higher consumption of vitamin A-rich rich plant source foods (39%), compared to those residing in residential food environments where these foods were not available (*p*-value < 0.01). The difference in the children’s consumption of vitamin A-rich plant source food between children from residentials with and without the availability of vitamin A-rich vegetables and roots, and dark green leafy vegetables was small (Table 3).

### 3.5. Consumption of Plant-Based Vitamin A-Rich Foods among Children by Household Wealth and Maternal Education

Children from the highest wealth quantile had significantly higher consumption of at least one source of vitamin A-rich plant source food (51%) than children from the lowest wealth quintile (25%) (*p*-value < 0.01). A similar pattern was observed in relation to maternal education: children whose mothers had college level education had a significantly higher consumption of at least one vitamin A-rich plant source food (48%) than children with lower maternal education (23%) (*p*-value < 0.01). The consumption pattern showed a marked improvement as the wealth status of the household improves and maternal education status increases (Table 3).

### 3.6. Multilevel Logistic Regression Analysis Results

In the full model (Model III), the odds of consuming plant-based vitamin A-rich food was 19% higher among children residing in residential food environments with vitamin A-rich fruits available, compared to those residing in residential food environments without this food type available (AOR: 1.20; 95% CI: 1.02, 1.41). However, the strength of this positive association was relatively weak when compared to the individual level influences on child consumption.

Children’s consumption of vitamin A-rich food increased progressively with maternal education and household wealth. The odds of consuming plant-based vitamin A-rich food were 2.55 times higher for children whose mothers were college-educated, compared to those children whose mothers had no education (AOR: 2.55; 95% CI: 2.01, 3.25). Similarly, the odds of consuming plant-based vitamin A-rich food were 2.37 times higher for children from households in the highest wealth quintile compared to children from households in the lowest quintile (AOR: 2.37; 95% CI: 1.92, 2.93) (Table 4). The association between residential availability and children consumption of vitamin A-rich food had no significant change after adjusting for individual-level factors. Overall, factors at the residential and individual level did not have a major influence on each other.

## 4. Discussion

The results of this study showed that only 36% of children had consumed plant-based vitamin A-rich food groups in the last 24 h during the survey. While residential availability of vitamin A-rich vegetables, roots, and dark green leafy vegetables had no association, the availability of vitamin A-rich fruits had a weak positive association with the children’s consumption of plant-based vitamin A-rich food. Higher household wealth and maternal educational status showed strong association with the consumption of plant-based vitamin A-rich food.

We found the overall consumption of plant-based vitamin A-rich food to be low among children in Addis Ababa. In particular, the intake of vitamin A-rich fruit among children was low (<10% of the children) compared to vitamin A-rich vegetables and roots and dark green leafy vegetables, as reported by previous research. This finding supports several existing studies in Ethiopia that also supported low intake of vitamin A-rich foods; for example, a study in 736 children in Sinan woreda, northwest Ethiopia, found that 15% of children were consuming vitamin A-rich fruits and vegetables [46]. Another study in Kachabira woreda, southern Ethiopia, found that, among 636 children, 18.8% were consuming vitamin A-rich fruits and vegetables [47]. This finding also supports wider trends of declining fruit and vegetable consumption in some countries in SSA over the past three decades, contributing to evidence from Angola, Madagascar, Botswana, Cote d’Ivoire, Guinea, Gabon, Uganda, and Rwanda [48]. Consistent with this finding, a multi-country analysis on fruits and vegetable consumption also reported low consumption of vitamin A-rich foods in low-income countries [49]. The low consumption could be explained by the purchasing ability of households, as these are more expensive for those in lower wealth quintiles [18]. Food preferences and traditional eating habits also influence food consumption habits. In Ethiopia, most families eat the staple diet that mainly consists of grains and legumes [50,51]; in this sense, a previous study also showed little difference in household dietary diversity even by wealth status except in the consumption of meat, fish, and fruits [19,35].

In this study, residential availability of plant-based vitamin A-rich items had no marked association with consumption, only a small effect of availability of vitamin A-rich fruits. This is contrary to the previous cross-sectional studies that found an association between availability fruits and vegetables in residential food environment and consumption [52,53,54]. In addition, a recent systematic review that indicated moderate evidence found an association between availability of fruit and vegetables in the retail food environment and consumption in LMICs [55]. Some potential explanations for the limited association between residential availability and consumption of plant-based vitamin A-rich food groups in our study could be related to the overall low consumption. In addition, mothers/caregivers may choose foods for children consumption based on the context of their own and household’s preferences and available resources (for example, a refrigerator is an essential resource for storing perishable foods such as fruit and vegetables) [56]. In our context, purchasing ability and socio-economic status may appears higher influence in the consumption than availability in the residential area. Expensive food items, even if they are available in the residential food environment, might not be purchased and consumed in families with economic constraints [57]. Lower income households are less likely to purchase recommended vegetables and other healthy foods [58]. Such consumption patterns eventually limit availability and drive prices up, as vendors shape their business based on their clients’ demand [59,60]. Another factor in our setting could be related to children’s preference of certain food items, such as fast food, which busy parents may easily give in to [51]. Yet another factor that perhaps obscure the association could be the family food practice. In Ethiopia, food is prepared for the whole family together, and as such, the quantity of food that satisfies the whole family is a higher priority than diversified and quality food [61].

In this study, maternal education was positively and strongly associated with children consumption of plant-based vitamin A-rich diet. Higher educational attainment is known to be an important factor for attitudes towards the consumption of the vitamin A-rich plant sources food groups [25,26,62,63]. Educated mothers are more likely to have nutrition literacy and healthier dietary practices, allowing them to mediate healthier diets for their children [64]. This could potentially be due to their higher exposure to media, as well as their ability to comprehend/filter the information they receive and make an informed decision [65]. Another potential explanation could be that educated mothers are better positioned to make decisions which foods to purchase for her children [66].

Children from the higher wealth status households had higher odds of consuming plant-based vitamin A-rich food. Household wealth status as measured in this study reflects on cumulative assets and resources [67] and could be a good indicator for purchasing ability of the family. In the study city, fruits and vegetable are relatively expensive, thus they are beyond the purchasing capability of the poorest households [18]. Evidence from the wider study indicated most households in the lowest wealth quantile perceived the consumption of vitamin A-rich fruits and vegetables as unaffordable [19]. Similarly, a study in Ghana also showed that children from poorer households were less likely to consume fruits and vitamin A-rich vegetables than children from the richest households [25]. These facts indicate the recommended fruit and vegetable consumption amount per day is generally unaffordable for most people in low-income countries [49], and most low-income people worldwide [68]. A FAO/WHO report stated that the difference in the amount of fruits and vegetables intake between the lowest and highest income quintiles households range from two-fold in Mozambique to eight-fold in Burundi [69]. Consumption of vegetables in the poorest wealth quintile is generally very low in sub-Saharan Africa [69]. On the other hand, fruit and vegetable intake could be greater in children in lower socio-economic positions due to local/traditional food norms that favor their consumption [27,28].

Although we were not able to explore social norms related to food consumption in this study, there is a growing recognition of social norms as an important factor influencing food consumption in low-income settings [70]. Social norms are unwritten rules that often influence child feeding behaviors. Social norms are likely to diffuse through social networks to reinforce local food consumption practices [71]. Social norm-based interventions were shown to improve food choices [71]. The use of theory in assessing the factors influencing food consumption patterns is uncommon and needs to be seriously considered in future research [72]. Although understanding relationships of the different levels of food information including the amount available per capita in the local food environment is critical, this kind of information is difficult to collect in low-income countries [73].

The strengths of the study include the inclusion of a large sample of preschool children and using observational checklist to assess the local residential food environment [74], as recommended for food environment studies in low-income country [43]. Another very important strength is related to the inclusion of all districts of the city in the surveys. Limitations of this study include the use of self-reported dietary intake of vitamin A-rich plant sources, which might be prone to minimize recall bias, although we used photo gallery-based standardized tools and experienced data collectors to minimize the bias. Additionally, we have not considered all factors that potentially influence consumption such as food price, purchasing power of household, nutrition knowledge, religious practices, food culture/beliefs, and cooking practices.

## 5. Conclusions

The consumption of plant-based vitamin A-rich food groups was low in Addis Ababa and had a positive but weak association with residential availability of vitamin A-rich fruits in the residential food environment. Consumption of plant-based Vitamin A-rich food was strongly associated with household wealth and maternal education. Further mixed method research is highly recommended to fully understand food consumption in children; qualitative research is required to understand the culture, belief, and preferences, and prospective quantitative for precisely measuring accessibility and consumption.

## Figures and Tables

**Figure 1 nutrients-14-00296-f001:**
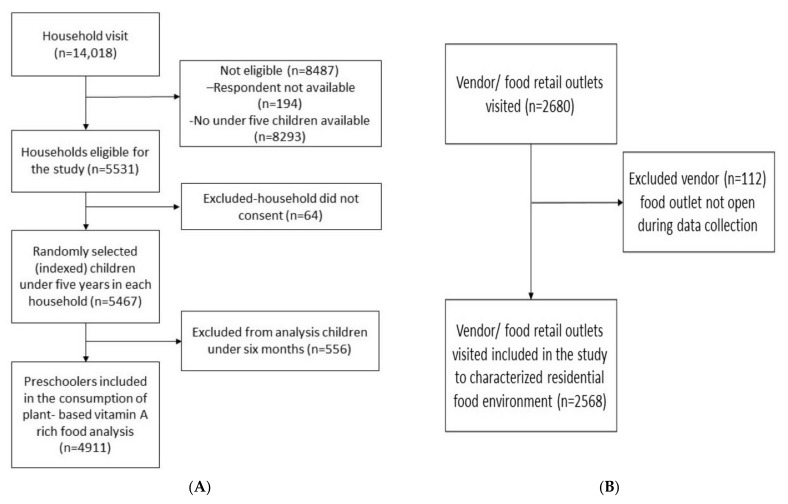
(**A**) Flow chart of households and children included in the study; (**B**) flow chart of vendors/food retail outlets included in the study.

**Table 1 nutrients-14-00296-t001:** Characteristics of household, mothers, and children (6–59months) (n = 4911) in Addis Ababa.

Level	Characterstics	n	%
Child	Sex		
	Male	2571	52.35
	Female	2340	47.65
	Age		
	6–23 months	1994	40.60
	24–59 months	2917	59.40
Women	Age		
	15–24	740	15.07
	25–34	2996	61.00
	35–44	927	18.88
	45 and above	248	5.05
	Married/living together	4298	87.52
	Education		
	Never finished grade	696	14.17
	Grade 1–4	443	9.02
	Grade 5–8	1466	29.85
	Grade 9–12	1355	27.60
	College	951	19.36
	Currently involved in any income earning activity	1290	26.27
Household	Household Food security status		
	Food secure	3003	61.15
	Mildly food insecure	457	9.30
	Moderately food insecure	963	19.61
	Severely food insecure	488	9.94
	Household Asset		
	Privately own house	1070	21.79
	Refrigerator	2697	54.92
	Car	382	7.78
	Improved drinking water source	4834	95.43
	Private toilet facility	1110	22.60

**Table 2 nutrients-14-00296-t002:** Residential availability of plant-based vitamin A-rich food groups in Addis Ababa (n = 233).

Type of Vitamin A-Rich Plant Source Foods	Residential Availability n (%)
Vitamin A-rich vegetables and roots	171 (73.39)
Dark green leafy vegetables	136 (58.37)
Vitamin A-rich fruits	84 (36.05)
At least one of the above ^a^	182 (78.11)

^a^ indicates the availability of vitamin A-rich vegetables and roots, vitamin A-rich fruit or dark green leafy vegetables in the residential food environment.

**Table 3 nutrients-14-00296-t003:** Children consumption of vitamin A-rich plant sources food by residential availability of any plant-based vitamin A-rich food and socio-economic status (N = 4911) in Addis Ababa.

Characteristics	Consumption % (CI)			
	Vitamin A-Rich Vegetables and Roots	Dark Green Leafy Vegetables	Vitamin A-Rich Fruits	Either of the Three Food Types
Residential Availability				
Vit A Veg				
Yes	20.84 (19.52–22.16)	N/A	N/A	35.71 (34.15–37.27)
No	19.26 (17.11–21.41)	N/A	N/A	34.96 (32.36–37.56)
*p*-value	0.23	N/A	N/A	0.67
DGL Veg				
Yes	N/A	17.24 (15.85–18.63)	N/A	36.81 (35.03–38.58)
No	N/A	15.71 (14.14–17.28)	N/A	33.73 (31.70–35.77)
*p*-value	N/A	0.15	N/A	0.03
Vit A Fruit				
Yes	N/A	N/A	10.18 (8.75–11.61)	38.86 (36.55–41.17)
No	N/A	N/A	7.83 (6.90–8.76)	33.71 (32.07–35.35)
*p*-value	N/A	N/A	<0.01	<0.001
Maternal Education Status				
Never attend/Not finished first grade	11.06 (8.73–13.40)	12.36 (9.91–14.80)	3.02 (1.75–4.29)	22.56 (19.45–25.67)
Grade 1–4	14.00 (10.76–17.23)	11.74 (8.74–14.74)	4.97 (2.94–6.99)	26.19 (22.09–30.28)
Grade 5–8	16.92 (15.00–18.84)	15.89 (14.02–17.77)	5.80 (4.60–7.00)	31.38 (29.00–33.75)
Grade 9–12	24.94 (22.64–27.25)	17.12 (15.11–19.13)	10.85 (9.19–12.51)	41.03 (38.41–43.65)
College	29.23 (26.34–32.13)	22.29 (19.65–24.94)	15.77 (13.45–18.09)	47.84 (44.67–51.02)
*p*-value	<0.001	<0.001	<0.001	<0.001
Household Wealth Status				
Lowest	14.49 (12.28–16.70)	11.22 (9.25–13.20)	4.59 (3.28–5.90)	25.10 (22.39–27.82)
Second	14.03 (11.86–16.22)	13.11 (11.00–15.23)	5.94 (4.46–7.43)	27.97 (25.15–30.79)
Middle	20.87 (18.33–23.42)	16.90 (14.56–19.25)	7.33 (5.70–8.96)	36.25 (33.24–39.26)
Fourth	20.79 (18.26–23.33)	16.94 (14.60–19.28)	9.33 (7.51–11.15)	36.71 (33.70–39.72)
Highest	31.81 (28.91–34.72)	24.72 (22.03–27.41)	16.01 (13.72–18.30)	51.37 (48.25–54.49)
*p*-value	<0.001	<0.001	<0.001	<0.001

A Chi-squared test was utilized to find the differences of children consumption by maternal education, household wealth, and residential availability of the particular food types. Abbreviation: N/A: Not Applicable, Vit A veg: Vitamin A-rich vegetables and roots; DGL veg: Dark green leafy vegetables; Vit A fruit: Vitamin A-rich fruits; either of the three food types— at least one of the plant-based vitamin A-rich food groups consumption.

**Table 4 nutrients-14-00296-t004:** Unadjusted and adjusted multilevel logistic regression result of the predictors of any plant-based vitamin A-rich food consumption in children aged 6 to 59 months (N = 4911).

Predictors	Any Plant-Based Vitamin A-Rich Food Intake			
	Unadjusted Model COR (95% CI)	Adjusted Model OR (95% CI)		
		Model I	Model II	Model III
Residential Availability (Yes/No)				
Vit A veg.	1.04 (0.87, 1.24)	0.87 (0.70, 1.07)		0.85 (0.70, 1.04)
DGL veg.	1.16 (0.99, 1.36)	1.15 (0.96, 1.40)		1.16 (0.97, 1.40)
Vit A fruit	1.27 (1.08, 1.49)	1.28 (1.08, 1.51)		1.20 (1.02, 1.41)
Maternal Educational Status				
Never attend	Ref.		Ref.	Ref.
Grade 1–4	1.19 (0.90, 1.58)		1.26 (0.95, 1.68)	1.24 (0.93, 1.66)
Grade 5–8	1.55 (1.25, 1.92)		1.60 (1.28, 2.00)	1.58 (1.26, 1.97)
Grade 9–12	2.33 (1.88, 2.88)		2.19 (1.74, 2.74)	2.12 (1.69, 2.66)
College	3.08 (2.46, 3.86)		2.62 (2.06, 3.33)	2.55 (2.01, 3.25)
Household Wealth Quintiles				
Lowest	Ref.		Ref.	Ref.
Second	1.15 (0.94, 1.41)		1.07 (0.87, 1.32)	1.06 (0.86, 1.31)
Middle	1.70 (1.39, 2.07)		1.46 (1.19, 1.79)	1.44 (1.18, 1.77)
Fourth	1.72 (1.40, 2.09)		1.46 (1.19, 1.79)	1.42 (1.16, 1.75)
highest	3.11 (2.55, 3.79)		2.44 (1.98, 3.00)	2.37 (1.92, 2.93)

Note: The reference to residential availability predictors (No), AOR: the multivariable multilevel regression model results in an adjusted odds ratio. COR: Bivariable multilevel regression results in an unadjusted odds ratio. Final model (III) LR test vs. logistic model: chibar2 (01) = 7.75 Prob ≥ chibar2 = 0.003. Model I adjusted for residential wealth status, household distance to indexed household. Model II adjusted for maternal age, child age, marital status, and number of under five children in the household. The final model (model III) adjusted for residential wealth, household distance to the index household, maternal age, child age, marital status, and number of under five children in the household. Vit A veg: Vitamin A-rich vegetables and roots; DGL veg: Dark green leafy vegetables; Vit A fruit: Vitamin A-rich fruits.

## Data Availability

Not applicable.

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
