# Peer review of "Residential Food Environment, Household Wealth and Maternal Education Association to Preschoolers’ Consumption of Plant-Based Vitamin A-Rich Foods: The EAT Addis Survey in Addis Ababa"

_nutrients, 2022, doi:10.3390/nu14020296_

Round 1

Reviewer 1 Report

The manuscript submitted for publication to Nutrients by Kebede et al., titled: “Residential food environment, household wealth and maternal education association to preschoolers’ consumption of plant-based vitamin A-rich foods: the EAT Addis survey in AddisAbaba” is an observational piece of work whereby the association of plant-based vitA consumption for preschoolers is assessed against factors including food environment, household wealth and maternal education. The topic is interesting in terms of providing potential factors for targeting malnutrition more effectively. While the overarching approach is not particularly novel, there is interested in this work presented herein due to the setting and the idiosyncrasies of the population especially when taking into consideration that the population/setting in discussion are understudied.

The reviewer would like to offer the following points for consideration:

  1. There are parts of the manuscript's narrative/text that are in different font and/or highlighted for no apparent reason.
  2. The introduction section is rather short while lacking statistics. It would be helpful to use statistical figures regarding the prevalence of VAD and or associated conditions based on either national statistics, and/or WHO, or available literature. 
  3. It would also be interesting to compare and contrast briefly how this situation is relative to the world average, the SubSaharan Africa average etc.
  4. What was the research question and the Ho hypothesis?
  5. Were the used questionnaires validated?
  6. Can the authors provide more detail and/or a reference for the data collection measurements?
  7. What were the inclusion and exclusion criteria for the participation in the study?
  8. Please try to provide a more crisp image of figure 1.
  9. Table 3 is very cluttered please consider revising it.
  10. In the discussion could the authors provide a theoretical framework as per the explanation of their findings?
  11. Please specify the authors contribution.

Reviewer 2 Report

This resubmitted and revised manuscript meets all my previous comments.

Round 2

Reviewer 1 Report

The authors have made a reasonable effort to address reviewer's comments.

This manuscript is a resubmission of an earlier submission. The following is a list of the peer review reports and author responses from that submission.

Round 1

Reviewer 1 Report

The novelty of research findings represents the major issue concerning this paper.

In the introduction section (lines 65 to 72) authors stated:

“In Ethiopia, the consumption of vitamin A-rich fruits and vegetables is very low in children [16]… The factors for low intake include household poverty, lower maternal education [18-20], and poor availability and affordability of vitamin A source food groups[4, 21]. The neighborhood food environment is also recognized as an important factor the influence food consumption globally [22], what is accessible and available in the neighborhood food environment influences the eating behaviors of families and their children [23].

These background data which justified the rationale of the study are almost the same as the main results and conclusions described in the Abstract  (line 41 to 44):

“This study showed only about a third of preschool children consumed plant-based vitamin A-rich food and consumption was influenced by maternal educational status, household wealth, and to a lesser extent by neighborhood availability of vitamin A-rich food groups.”

as well as in the Conclusions section (line 341 to 344):

“The consumption of plant-based vitamin A-rich food groups was low in Addis Ababa and not highly influenced by neighborhood availability of vitamin A-rich plant source food groups. However, household wealth and maternal education strongly predicted children's consumption of plant-based vitamin A-rich foods.”

Therefore authors should prove (e.g. in the Discussion) the high novelty of their research.

Reviewer 2 Report

The manuscript submitted for publication to Nutrients by Kebede et al., titled: “Residential neighborhood availability, household wealth and maternal education association to preschoolers’ consumption of plant-based vitamin A-rich foods: the EAT-Addis survey in Addis Ababa” aimed to investigate the association between plant based richly food sourced vitamin A consumption by preschoolers and availability, household wealth and maternal education in Addis Ababa, Ethiopia.

The reviewer presents some points for consideration by the authors for the improvement of the manuscript according to his professional opinion.

1. The introduction is rather short and limited in terms of its referencing, while block referencing is observed. It is recommended that the introduction includes more information about the topic as it relates to Ethiopia, provide global and local statistics and percentages thus providing concrete facts, rather than making statements with block referencing.

2. In the introduction it is important to clearly define the research question and the hypothesis of the authors.

3. Please clarify the contribution to the field and the novelty of the study in the introduction.

4. When access to vitamin A rich foods is considered it is important to account for price, price/minimum wage, purchasing power of household. Were these factors considered in the analysis by the authors?

5. Another factor for consideration it terms of access to vitamin A rich foods is the actual amount per capita in addition to. physical location. In other words the availability or whether there is enough product and vitamin A equivalent units per capita and how those compare to dietary recommendations.

6. Were the questionnaires used in the study validated for the population assessed?

8. What were the inclusion and exclusion criteria for the participants in the study? The authors discuss in detail the household selection mode but how about criteria pertinent to the actual family members?

9. Were confounding factors accounted for and corrected for (Eg. age, beliefs, knowledge, attitude, practices, nutrition knowledge, religious practices, food culture, cooking practices etc).

10. A 24h recall may not be optimum to determine the main outcome variable. Why the authors did not use a 3 day food record instead?

11. Discussion section is short and relatively low referenced. The finding that maternal education level and SES is positively associated with children consumption of vitamin A rich plant-based foods is not particularly surprising. In this sense what is the novelty and the uniqueness of the work? The reviewer understands that the setting is understudied, but the discussion section is the appropriate place to elaborate on a scientific discussion with strong review of the literature for support, whereby the idiosyncrasies and uniqueness of the setting are dissected towards explaining the results and findings along with employment of behavioral theories. The reviewer, believes in his professional opinion, that this is the type of discussion section that would serve the manuscript and the rest of the scientific community the most and recommends that such an effort is undertaken by the authors to upgrade the discussion section in that regard. The manuscript would thus become more informational to a wider readership in Nutrition.

Round 2

Reviewer 1 Report

I carefully read the revised manuscript and I am satisfied with authors corrections and response.

Reviewer 2 Report

The authors have made a reasonable effort in addressing the reviewer's comments.